# Invariant representation of physical stability in the human brain

**RT Pramod[1,2]\*, Michael A Cohen[2,3], Joshua B Tenenbaum[1,2], Nancy Kanwisher[1,2]**

[1]Center for Brains, Minds and Machines, Massachusetts Institute of Technology, Cambridge, United States; [2]Department of Brain and Cognitive Sciences, Massachusetts Institute of Technology, Cambridge, United States; [3]Amherst College, Amherst, United States

**Abstract** Successful engagement with the world requires the ability to predict what will happen next. Here, we investigate how the brain makes a fundamental prediction about the physical world: whether the situation in front of us is stable, and hence likely to stay the same, or unstable, and hence likely to change in the immediate future. Specifically, we ask if judgments of stability can be supported by the kinds of representations that have proven to be highly effective at visual object recognition in both machines and brains, or instead if the ability to determine the physical stability of natural scenes may require generative algorithms that simulate the physics of the world. To find out, we measured responses in both convolutional neural networks (CNNs) and the brain (using fMRI) to natural images of physically stable versus unstable scenarios. We find no evidence for generalizable representations of physical stability in either standard CNNs trained on visual object and scene classification (ImageNet), or in the human ventral visual pathway, which has long been implicated in the same process. However, in frontoparietal regions previously implicated in intuitive physical reasoning we find both scenario-invariant representations of physical stability, and higher univariate responses to unstable than stable scenes. These results demonstrate abstract representations of physical stability in the dorsal but not ventral pathway, consistent with the hypothesis that the computations underlying stability entail not just pattern classification but forward physical simulation.

**\*For correspondence:**
pramodrt@mit.edu

**Competing interest:** The authors declare that no competing interests exist.

## Editor's evaluation

This is an intriguing study using cleverly designed stimuli to investigate the representation of physical stability in the human brain. This paper will be of interest to readers wondering when human cognition uses pattern matching similar to that used by machine learning algorithms, and when it relies on more specialised processes evolved for specific tasks. The well-crafted experiments provide convincing support for the authors' claim that the human brain computes an abstract representation of the physical stability of visual scenes.

## Introduction

Visual percepts are imbued with possibility. We see not just a dog pointing motionless at a rabbit, but a dog about to give chase; not just a wineglass near the edge of a table but a wineglass about to smash on the floor; not just a cup filled to the brim with coffee, but a cup at risk of staining the white tablecloth beneath. This ability to see not only the current situation in front of us, but what might happen next, is essential for planning any action. Although some of our predictions concern other people and what they will think and do, many of our predictions concern the physical world around us. A fundamental prediction we make about the physical world is whether it is stable, and hence unlikely to change in the near future, or unstable, and likely to change. Here, we ask whether the brain

computes physical stability via simple pattern classification, such as that thought to be conducted in the ventral visual pathway, or whether instead we determine stability by running a kind of simulation in our head to determine what, if anything, will happen next.

What kind of representations might be required for determining the physical stability of a scene? Computational work has shown that convolutional neural networks (CNNs) trained on ImageNet learn powerful representations that support not only object recognition, but multiple other tasks like object detection, segmentation, and retrieval (*Liu et al., 2015*; *Golub et al., 2020*; *Babenko et al., 2014*; *Razavian et al., 2014*). Indeed, several studies have claimed that these networks can accurately predict scene stability (*Lerer et al., 2016*; *Conwell et al., 2019*; *Li et al., 2016*). But CNNs have so far been tested only within very narrow domains (e.g., block towers with several cubes of different colors, in *Figure 1A*, left column) and it is not clear they would successfully predict future states of complex natural images (e.g., *Figure 1A*, middle two columns). A *general* ability to predict what will happen next in physical scenarios may instead require a richer representational of the physical properties of the scene that support forward simulations like those in the physics engines of video games (*Battaglia et al., 2013*; *Ullman et al., 2017*). Behavioral evidence from humans is conflicting here. On the one hand, human observers can perceive the stability extremely rapidly and preattentively (*Firestone and Scholl, 2017*; *Yang and Wolfe, 2020*), and these findings have been taken to argue for a simple feedforward rather than a simulation process. On the other hand, human judgments of the stability of a tower of blocks are well modeled by approximate probabilistic simulation in game-style physics engines (*Battaglia et al., 2013*; *Ullman et al., 2017*; *Zhang et al., 2016*; *Gerstenberg et al., 2017*), which better fit the human pattern of errors than CNNs do (*Zhang et al., 2016*). Perhaps most importantly, humans can easily judge the physical stability of a wide variety of novel real-world images, which is likely to require a more abstract and generalizable representation of physical properties of the scene. Here, we test whether representations that support fast feedforward visual object and scene recognition in machines (i.e., CNNs trained on ImageNet) and brains (i.e., the ventral visual pathway) can support discrimination of the physical stability of complex natural scenes, or whether different kinds of computations – in particular, those that explicitly model dynamic physical properties of the scene, perhaps through a kind of simulation mechanism – may be required to determine physical stability in both machines and brains.

To investigate this question, we first curated a set of natural images depicting real-world scenarios of objects in physically stable versus unstable configurations (the 'Physical-Objects' set, *Figure 1A*), as well as a generalization set of images of people in physically stable versus unstable situations (the 'Physical-People' set, *Figure 1A*). We then presented these stimuli to CNNs pretrained on ImageNet classification and asked if the resulting representations are sufficient to discriminate the physical stability of the scene in a fashion that generalizes across image sets. Next, we presented the same to human participants being scanned with fMRI and asked the same question of the resulting neural responses in two key brain regions. One such region is the ventral visual pathway, which is widely thought to conduct the core computations underlying visual object recognition, and which is well modeled by CNNs trained on ImageNet (*Yamins et al., 2014*; *Khaligh-Razavi and Kriegeskorte, 2014*). Thus, if representations optimized for object recognition suffice for determining the physical stability of novel scenes, we would expect to find this ability in the ventral visual pathway. However, an alternative hypothesis is that this information will be found instead in a set of parietal and frontal regions that have been shown to be engaged when people predict what will happen next in physical (more than social) scenarios (*Fischer et al., 2016*), and hold information about physical variables such as mass, invariant to the scenario that revealed that mass (*Schwettmann et al., 2019*). We have hypothesized (*Fischer et al., 2016*) that these regions (referred to here as the 'Physics Network') may contain a generative model of dynamic physical scenes capable of forward simulation, which can be used for prediction, planning, and many other tasks. Thus, if determining the stability of a scene requires an abstract representation of the physical properties of the scene and the ability run forward simulations on those representations, then the Physics Network would be a more likely locus of these representations than the ventral visual pathway.

Finally, we further conjectured that if perception of physical stability is determined by the operation of a forward simulation mechanism akin to a generative physics engine, then we might find higher neural activity in the Physics Network for physically unstable than stable situations because there is more to simulate in an unstable situation. That is, simulations of unstable scenes where objects are

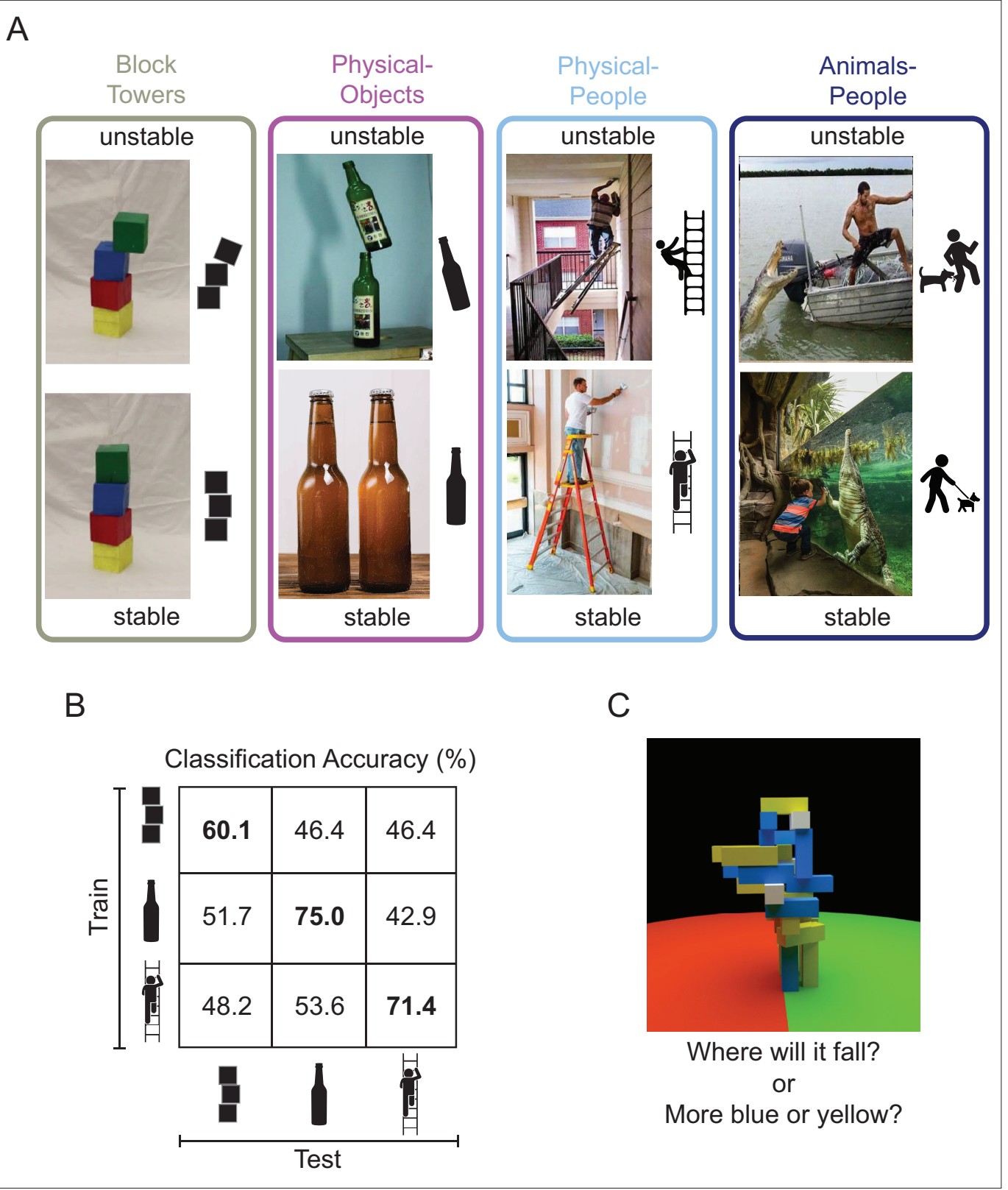

**Figure 1.** Design of studies conducted on machines and brains. (**A**) Image sets depicting (1) unstable versus stable block towers (*left*), (2) objects in unstable versus stable configurations (*middle left, the Physical-Objects set*), (3) people in physically unstable versus stable situations (*middle right, the Physical-People set*), and (4) people in perilous ('unstable') versus relatively safe ('stable') situations with respect to animals (*right, the Animals-People set*). Note that by 'physically unstable' we refer to situations that are currently unchanging, but that would likely change dramatically after a small

*Figure 1 continued on next page*

*Figure 1 continued*

perturbation like a nudge or a puff of wind. The former three scenarios were used to study the generalizability of feature representations in a deep convolutional neural network (CNN) and the latter three scenarios were used in the human fMRI experiment. (**B**) Accuracy of a linear support vector machine (SVM) classifier trained to distinguish between CNN feature patterns for unstable versus stable conditions, as a function of Test Set. Accuracy within a scenario (i.e., train and test on the same scenario) was computed using a fourfold cross-validation procedure. (**C**) A functional localizer task contrasted physical versus nonphysical judgments on visually identical movie stimuli (adapted from Figure 2A from *Fischer et al., 2016*). During the localizer task, subjects viewed movies of block towers from a camera viewpoint that panned 360° around the tower, and either answered 'where will it fall?' (physical) or 'does the tower have more blue or yellow blocks?' (nonphysical).

expected to move require the construction and updating of dynamic representations that unfold over time, whereas simulations of stable scenes, where objects are expected to remain stationary, do not require such dynamically updated representations. Thus, a higher univariate response to unstable than stable scenes would provide preliminary (albeit not definitive) evidence for forward simulation. Importantly, participants were never asked to judge or think about the physical stability of the scenario they were viewing, which enabled us to ask whether this information is extracted even when not required by the task (*Firestone and Scholl, 2016*).

## Results
### Feedforward CNNs trained on ImageNet do not have a generalizable representation of physical stability

Previous studies have shown that visual features extracted by feedforward deep CNNs (like AlexNet, *Ferrari et al., 2018*; VGG, *Li et al., 2016*; and ResNet, *Lerer et al., 2016*; *Conwell et al., 2019*) can distinguish between stable and unstable towers. Although these features produced above chance classification performance within a scenario (i.e., train on towers with two blocks and test on towers with four blocks), they have not been tested on very different scenarios and hence their ability to represent an abstract and generalizable notion of physical stability remains unknown. Significant generalizability of these features to novel image sets would provide some evidence that physical stability can be inferred from a simpler mechanism of visual feature extraction in pattern classification systems rather than running forward simulations of the kind used in game physics engines.

To test this question, we extracted features for our stable and unstable images from the final fully connected layer (*fc1000*, the layer immediately preceding the SoftMax probability layer) of ResNet-50 – a feedforward deep CNN trained on ImageNet dataset for object recognition (*He et al., 2016*). A shallower variant of this network has been previously used for stability classification (*Lerer et al., 2016*; *Conwell et al., 2019*). In addition to the complex real-world scenarios ('Physical-Objects' and 'Physical-People'), we also chose Block Towers to replicate previous findings. Within each scenario (Block Towers, Physical-Objects, and Physical-People, *Figure 1A*), we trained a linear support vector machine (SVM) classifier to distinguish between feature patterns for stable and unstable images and tested it on held-out images using fourfold cross-validation. These analyses replicated previous results by showing above chance classification accuracy within the Block Towers scenario and further found higher classification performance within our two new natural image sets (60.1%, 75%, and 71.4% for Block Towers, Physical-Objects, and Physical-People scenarios, respectively; diagonal cells in *Figure 1B*, $p < 0.05$ for a bootstrap test with $n = 10,000$). However, this performance did not generalize across scenarios: accuracy was near chance for all cross-validation tests in which the SVM classifier was trained on one scenario and tested on a different scenario (i.e., off-diagonal cells in *Figure 1B*, $p > 0.15$ for a bootstrap test with $n = 10,000$). Thus, visual features from this feedforward deep CNN do not carry abstract and generalizable information about physical stability. We also found similar results with a VGG-16 network trained on ImageNet. The fact that these CNN features can distinguish stability within a scenario but not across scenarios suggests that the discriminative features underlying this discrimination might be related to incidental properties that are indicative of stability only within a particular scenario (e.g., the displacement of each block from the centroid of the block below).

These results indicate that the ImageNet-trained CNNs previously claimed to support judgments of stability cannot do so in a fashion that generalizes to novel image sets. This result further suggests that brain regions thought to be optimized (through development and/or evolution) for similar tasks (ventral temporal cortex) might similarly fail to contain scenario-invariant information about physical

stability, and that the human ability to perform this task may result from different kinds of representations elsewhere in the brain, such as the frontoparietal cortical regions previously implicated in physical inference. We test these hypotheses next.

## A generalizable representation of physical stability in cortical regions previously implicated in intuitive physical inference, but not in the ventral temporal cortex

We first identified functional regions of interest (fROI) in each participant individually, including ventral temporal cortex (VTC) and the candidate Physics Network in the frontal and parietal lobes, by intersecting anatomical constraint parcels for these regions with the relevant functional activations (see *Figure 2A* and Methods). All participants had significant voxels within both VTC and physics parcels (number of voxels per participant, mean ± standard error of the mean [SEM] =350.5 ± 68.7 for VTC ROIs and mean ± SEM =311.3 ± 69.8 for physics ROIs). We further collected fMRI responses in these regions to stable and unstable photos of real-world scenarios (Physical-Objects and Physical-People) as well as an additional scenario type, 'Animals-People', depicting people in either perilous ('unstable') or nonperilous ('stable') conditions due to an animal rather than a physical object. We used the Animals-People scenarios as a control for potential confounds of peril, arousal or attention. Specifically, we predicted that representations of physical stability to be generalizable across Physical-Objects and Physical-People scenarios but not to Animals-People scenario.

### Within-scenario stability information

To test whether VTC and Physics Network contain information about physical stability, we measured the fMRI response of each voxel of the fROI in each participant, separately for odd and even runs in each of the six stimulus conditions of the main experiment. We then used multivoxel pattern analysis (*Haxby et al., 2001*) to test whether physical stability could be decoded in each fROI. Specifically, across all voxels in each fROI we computed the correlation between the response patterns *within* stability conditions (stable even to stable odd, and unstable even to unstable odd) and *between* stability conditions (stable even to unstable odd, and stable odd to unstable even) for each of the scenarios separately (see *Figure 2A* and Methods). For VTC, we found that the *within* condition pattern correlations were significantly higher than the *between* condition pattern correlations for both Physical-Object and Physical-People scenarios (see *Figure 3* and *Figure 2—figure supplement 1A*), indicating the presence of information about stability for each scenario type. However, pattern information did not distinguish perilous (or 'unstable') from nonperilous ('stable') situations in the Animals-People scenarios (see *Figure 3* for stats). Further, this difference in stability information across scenario types was itself significant, as revealed by a significant interaction between-scenario type and within versus between correlations in an analysis of variance (ANOVA) ($F_{1,12}$ = 72.2, p = 0.000002). The Physics Network showed the same pattern of results (*Figures 2B and 3*), with significantly higher correlations within than between stability for both the Physical-Object and Physical-People scenarios but not the Animals-People scenarios, and a significant ANOVA interaction between-scenario type and within versus between correlations ($F_{1,12}$ = 22.9, p = 0.0004). Thus, both VTC and the Physics Network contain information about physical (but not animate) stability when analyzed within scenarios.

### Between-scenario stability information

However, our strong prediction was that any brain region that represents abstract physical information should show patterns of response that generalize across physical scenarios (Physical-Objects and Physical-People). To test this, we again used multivoxel pattern correlation analysis to compute *within*-condition (unstable–unstable and stable–stable) and *between*-condition (unstable–stable and vice versa) correlations, but this time across scenario types (*Figure 2C*). Indeed, we found that in the Physics Network (and not in VTC – see *Figure 2—figure supplement 1B*), the within condition pattern correlations are significantly higher than between condition pattern correlations only across Physical-Objects and Physical-People scenarios, but not across Physical and Animate scenarios (*Figures 2D and 3*). Further, we found a significant interaction between fROI (physics vs. VTC) and between-scenario stability information (within vs. between correlations), indicating stronger generalization across scenarios in the Physics Network compared to the VTC fROI ($F_{1,12}$ = 6.29, p = 0.027). Thus, the Physics Network (and *not* VTC) holds information about physical stability that generalizes

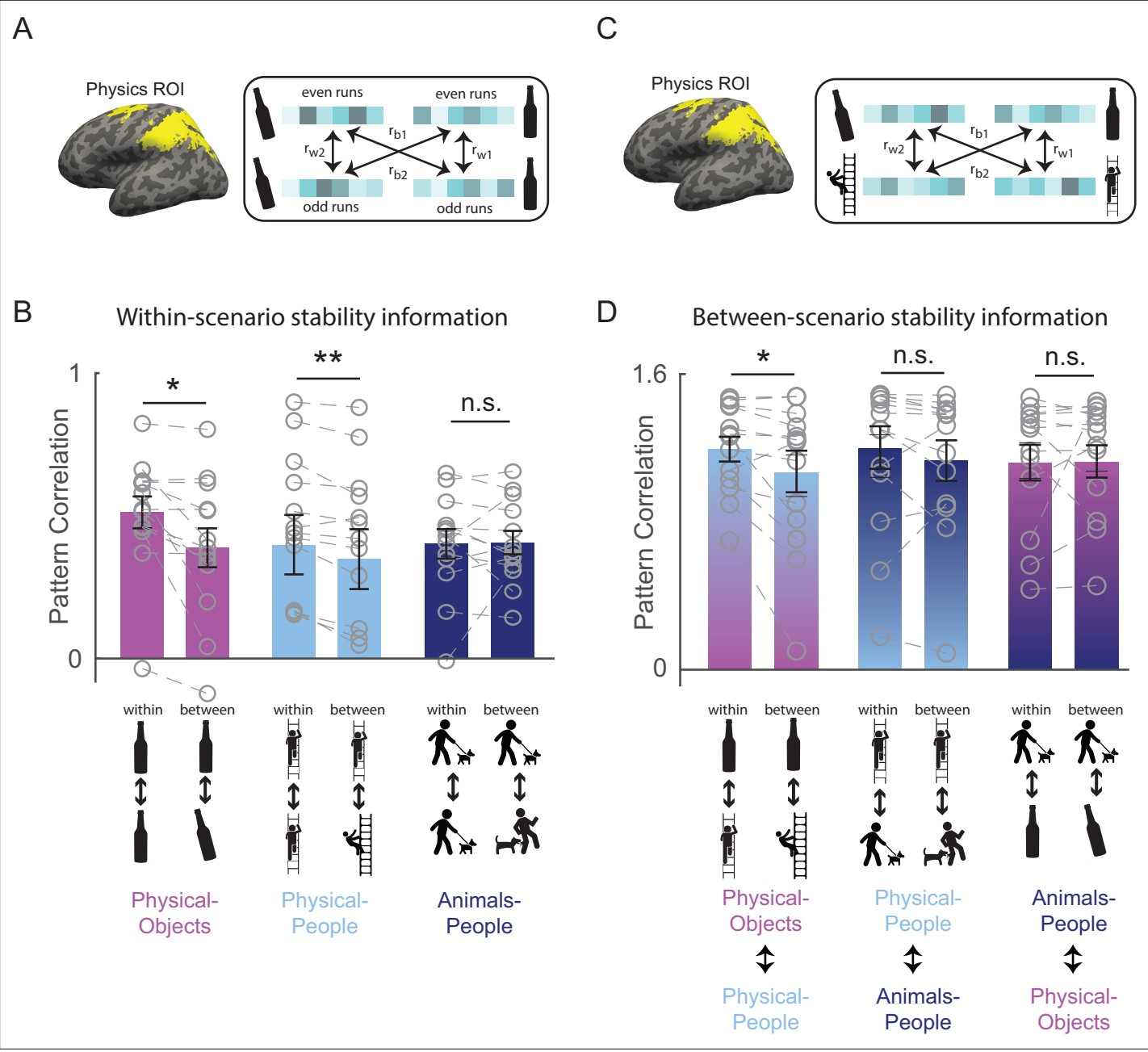

**Figure 2.** Generalizable representation of physical stability in the human brain. (**A**) Functionally defined regions of interest implicated in intuitive physical inference (the Physics Network) were defined in each participant individually by intersecting the activation from a localizer task with anatomical constraint parcels in the frontal and parietal lobes shown in yellow (see Methods). Patterns of activity across voxels in these functional regions of interest (fROIs) were extracted separately in each participant for each combination of even and odd runs, unstable and stable conditions, and scenario (Physical-Objects, Physical-People, and Animals-People). Following standard practice (*Haxby et al., 2001*), correlations between even and odd runs in the pattern of response across voxels were computed both within stability conditions ($r_{w1}$ = stable even to stable odd and $r_{w2}$ = unstable even to unstable odd), and between stability conditions ($r_{b1}$ = stable even to unstable odd and $r_{b2}$ = unstable even to stable odd), for each of the three scenario types. (**B**) Bar plot showing average pattern correlations *within* and *between* conditions with paired *t*-tests done after Fisher transformation, for each scenario type. (**C**) is similar to (**A**) but the within and between stability pattern correlations were computed across scenario types ($r_{w1}$ and $r_{w2}$ indicate pattern correlations computed within stable or *within* unstable conditions across scenario, and $r_{b1}$ and $r_{b2}$ indicate pattern correlations computed *between* stable and unstable conditions across scenario). (**D**) is same as in (**B**) but for pattern correlations computed across scenarios. Note that the across scenario correlations in (**D**) are overall higher than within-scenario correlations in (**C**) because the data were not split into odd and even runs for the across scenario case. Gray circles and the corresponding connecting lines denote individual subject's data. Error bars indicate standard error of mean across subjects. ** and * indicate statistically significant effect at $p \leq 0.005$ and $p \leq 0.05$, respectively, and n.s. indicates no statistically significant effect.

*Figure 2 continued on next page*

*Figure 2 continued*

The online version of this article includes the following figure supplement(s) for figure 2:

**Figure supplement 1.** Stability decoding in ventral temporal cortex (VTC) regions of interest (ROI).

from scenarios with only inanimate objects to scenarios in which people are in physically unstable situations, but not to scenarios in which people are in peril from animals.

## Eye movement and attention controls

Could these findings in the Physics Network simply be due to differential eye movements across different experimental conditions (despite the instructions to fixate), or differential attention? To explore these possibilities, we quantified eye movements collected in the scanner during the main experiment in a subset of participants (*n* = 6) for each of the six stimulus conditions by extracting the average *x* and *y* coordinates of eye position, as well as the number, duration, and amplitude of eye movements. None of these quantities showed a significant difference between stable and unstable conditions in any of the scenarios except for saccade amplitude in the 'Physical-People' scenario

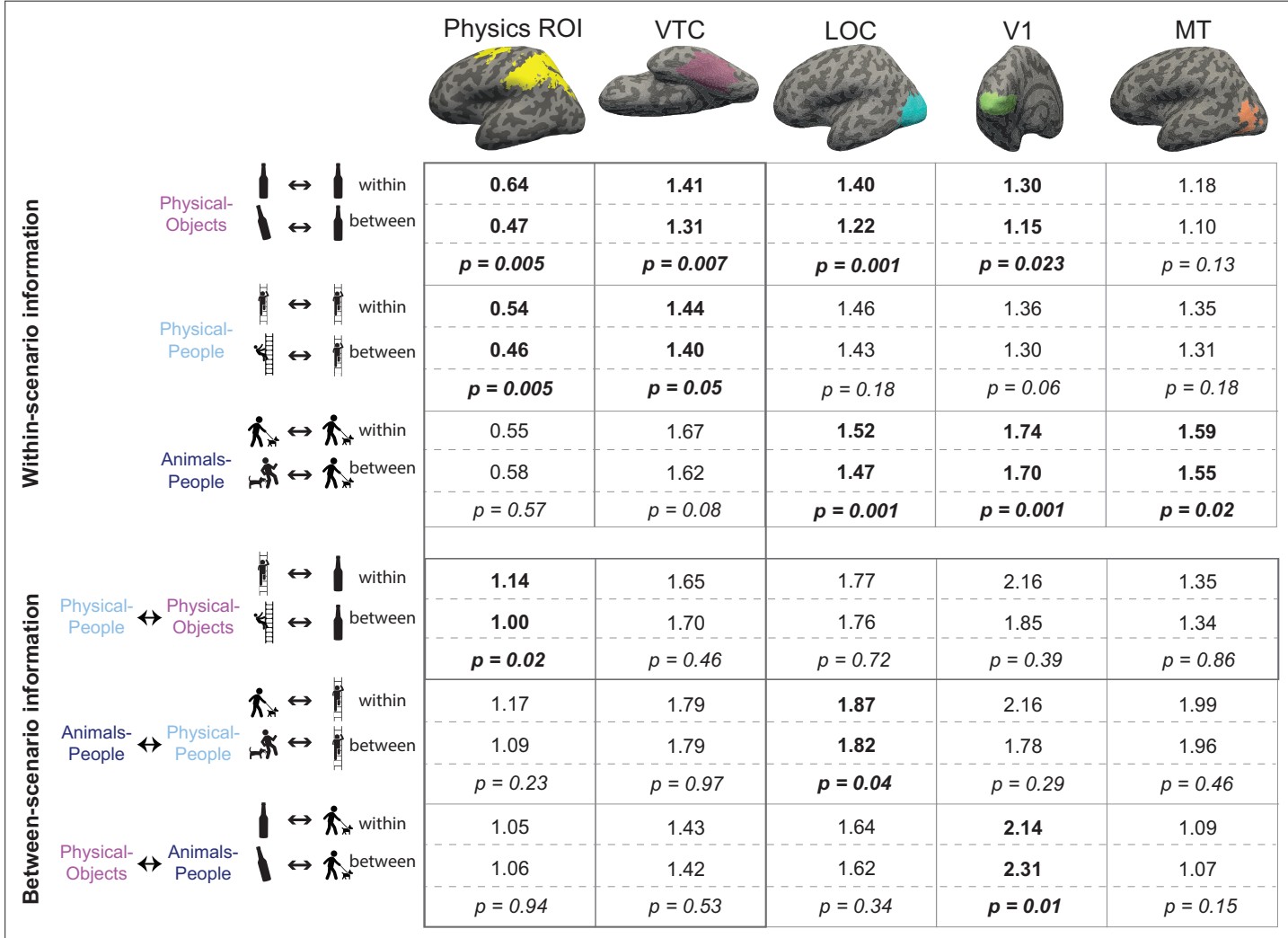

**Figure 3.** Multivoxel pattern correlation analysis in all regions of interest (ROIs). Each cell shows the average Fisher transformed correlation within and between stability conditions, along with the p value for a paired *t*-test comparing the two sets of values. Each column includes the results from one functional regions of interest (fROI). The top three rows contain results for the analyses within-scenario type and the bottom three rows show results for the pattern correlation analysis across scenarios. Significantly higher correlations for within than between conditions are highlighted in bold. The row depicting the crucial cross-decoding result with significant generalization only in the Physics Network ROI is highlighted with a darker bounding box.

(p = 0.028, *Supplementary file 1*). However, we found no significant interaction of stability with scenario type for any of the eye movement quantities using separate ANOVAs (p > 0.1). Thus, the observed differences in brain activations for our stimulus conditions are unlikely to be explained by differences in retinal input patterns due to eye movements. Second, analysis of subjective ratings by a separate set of subjects (see Methods) of '*interestingness*' of our stimuli (which we used as a proxy for how attention-grabbing the stimuli were) revealed significantly higher ratings for unstable over stable conditions in all three scenarios (p < 0.001 for a paired *t*-test on average ratings across subjects; *Supplementary file 1*, last column). However, we found no interaction between stability and scenario type in the ANOVA (p = 0.23), implying that the difference in ratings between stability conditions did not vary between physical and animate scenarios.

Further, to confirm that these results are not due to systematic differences in the ease of understanding of the images, we ran a behavioral task (see *Supplementary file 6* on *Stability Detection Behavioral Experiment*) where subjects were asked to categorize the stability of images. We found that reaction times differed significantly between stable and unstable conditions within each scenario (p < 0.05), but the direction of the effect was not consistent across scenarios. Most importantly, the overall pattern of reaction time and accuracy was not consistent with an account of our fMRI results in terms of difficulty (see *Supplementary file 4* for reaction time and *Supplementary file 5* for accuracy data).

Thus, the distinctive and generalizable representation of physical stability we found in the Physics Network are unlikely to be due to differential eye movements, attention, or difficulty confounds.

## Analysis of subregions of the physics fROIs

The Physics Network considered here includes regions in both the parietal and frontal lobes in both hemispheres. Evidence that the scenario-invariant representation of physical stability reported above is not restricted to a subset of these regions comes from separate analyses of the left parietal, left frontal, right parietal, and right frontal fROIs in each participant. An ANOVA analyzing the generalization of stability information (between vs. within condition pattern correlations across Physical-Objects and Physical-People; $F_{1,12} = 6.11$, p = 0.03 for the main effect) did not find a significant interaction of stability information with either hemisphere (left vs. right; $F_{1,12} = 2.75$, p = 0.12) or lobe (parietal vs. frontal; $F_{1,12} = 0.24$, p = 0.63; *Supplementary file 2*). However, the stability information was stronger in the parietal compared with frontal lobe (*Supplementary file 2*).

## Other visual regions do not have a generalizable representation of physical stability

In the previous section, we showed that the Physics Network in the frontoparietal cortices, but not VTC, has a representation of physical stability that generalizes across scenario. Is this abstract representation also found in other visual regions? We functionally defined two other visual regions: V1 and lateral occipital complex (LOC) (see Methods) and performed multivoxel pattern correlation analyses as before. In both regions, we found significant decoding of stability only in Physical-Objects scenario but no cross-decoding of stability across physical scenarios (*Figure 3*). In addition, the interaction of this cross-decoding effect with fROI was significant in separate ANOVAs contrasting the Physics Network with V1 ($F_{1,12} = 6.31$, p = 0.027) and LOC ($F_{1,12} = 5.63$, p = 0.035). Thus, a generalizable representation of physical stability is apparently a distinctive property of the Physics Network in the parietal and frontal lobes and is not a widespread property of visual cortex.

The only other significant generalization we observed was for the Animals-People and Physical-People scenarios in LOC. This effect was not predicted, and could reflect either similar shape information being present across the two scenarios, or an abstract representation of peril/risk to people.

## Higher mean responses to physically unstable than stable scenes in physics fROI

Previous studies have proposed that human intuitive physical reasoning, including inferences about physical stability, can be explained by a model that performs probabilistic simulations of the future states of the physical world (*Battaglia et al., 2013*; *Zhang et al., 2016*). Do the candidate physics regions in our brain perform this forward simulation of what will happen next? Some evidence for this idea comes from the higher response of these regions during physical prediction than color-judgment

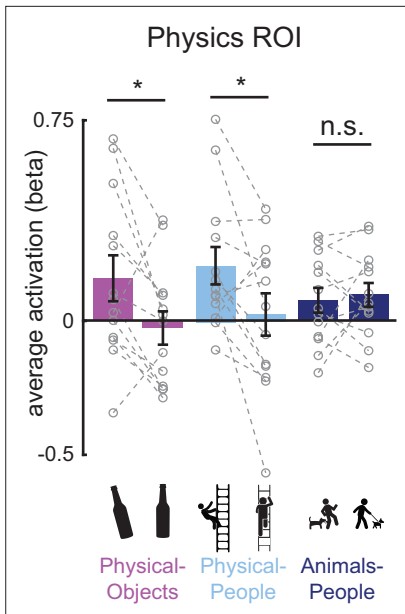

**Figure 4.** Bar plot showing the average activation (GLM beta estimates) in the physics functional regions of interest (fROI) for both stable and unstable conditions in each of the three scenario types (Physical-Objects, Physical-People, and Animals-People). Gray circles and the corresponding connecting lines indicate individual subject's data. Error bars indicate standard error of mean across subjects. * indicates significant effect at p < 0.05 and n.s. indicates no statistically significant effect.

tasks used in our localizer and in the original fMRI study that used this task (*Fischer et al., 2016*). However, neural activity in that contrast could simply reflect the process of building a mental model of the physical scene (including object properties and relationships), not predicting or simulating what would happen next. Here, we reasoned that if the candidate physics regions are engaged automatically in simulating what will happen next, they should show a higher mean response when viewing physically unstable scenes (because there is more to simulate) than stable scenes (where nothing is predicted to happen).

We tested this prediction by comparing the average fMRI response for unstable and stable conditions in each of the three scenario types, in the Physics Network (defined in the same way as in the previous analysis). As predicted, the physically unstable condition showed a significantly greater response compared to the physically stable condition in both 'Physical-Objects' (p = 0.02 for a paired *t*-test on average response across subjects) and 'Physical-People' scenarios (p = 0.04 for a paired *t*-test on average response across subjects; *Figures 4 and 5*, left column), but not the 'Animals-People' scenario (p = 0.66 for a paired *t*-test on average response across subjects; *Figures 4 and 5*). As before, we also checked whether this trend is largely driven by voxels in one of the hemispheres (right/left) or one of the lobes (parietal/frontal) by performing an ANOVA. We found a significant interaction between stability (unstable vs. stable) and lobe ($F_{1,12}$ = 8.28, p = 0.014) but not hemisphere ($F_{1,12}$ = 3.04, p = 0.11) for the 'Physical-Objects' scenarios, and the same pattern for the 'Physical-People' scenario (stability × lobe: $F_{1,12}$ = 8.53, p = 0.013; stability × hemisphere: $F_{1,12}$ = 0.34, p = 0.57). Post hoc analysis revealed that this interaction effect was largely driven by the univariate difference in the parietal but not frontal lobe (*Supplementary file 3*).

According to our hypothesis, it is the automatic simulation occurring in the Physics Network that leads to the higher response for unstable than stable conditions. As such we predicted that we would not see this same effect in VTC and other visual regions (V1 and LOC) that are not engaged in physical simulation. Indeed, as shown in *Figure 5*, we did not. Furthermore, we found a significant interaction of stability with region (physics fROI vs. visual fROI) for the 'Physical-Objects' ($F_{1,12}$ = 13.04, p = 0.0035 for V1; $F_{1,12}$ = 8.27, p = 0.014 for VTC) and the 'Physical-People' scenarios ($F_{1,12}$ = 11.14, p = 0.0059 for V1; $F_{1,12}$ = 7.83, p = 0.016 for LOC).

Thus, physically unstable scenes evoke stronger responses than stable scenes in the frontoparietal Physics Network, but not elsewhere, consistent with the hypothesis that these regions are engaged in running forward simulations of what will happen next.

## Higher responses to any instability in visual motion area MT

Finally, we predicted that visual motion area MT might show higher responses for all three forms of instability, whether physical (with objects or people), or animate. Our rationale was that because MT is implicated in motion processing in general, any forward simulation involving motion, whether physical or animate, could activate this region. Indeed, that is exactly what we saw. As shown in *Figures 3 and 5* (last column), we found significantly higher responses in MT for unstable/perilous than stable/safe scenes for all three types of scenarios without any significant stability information in

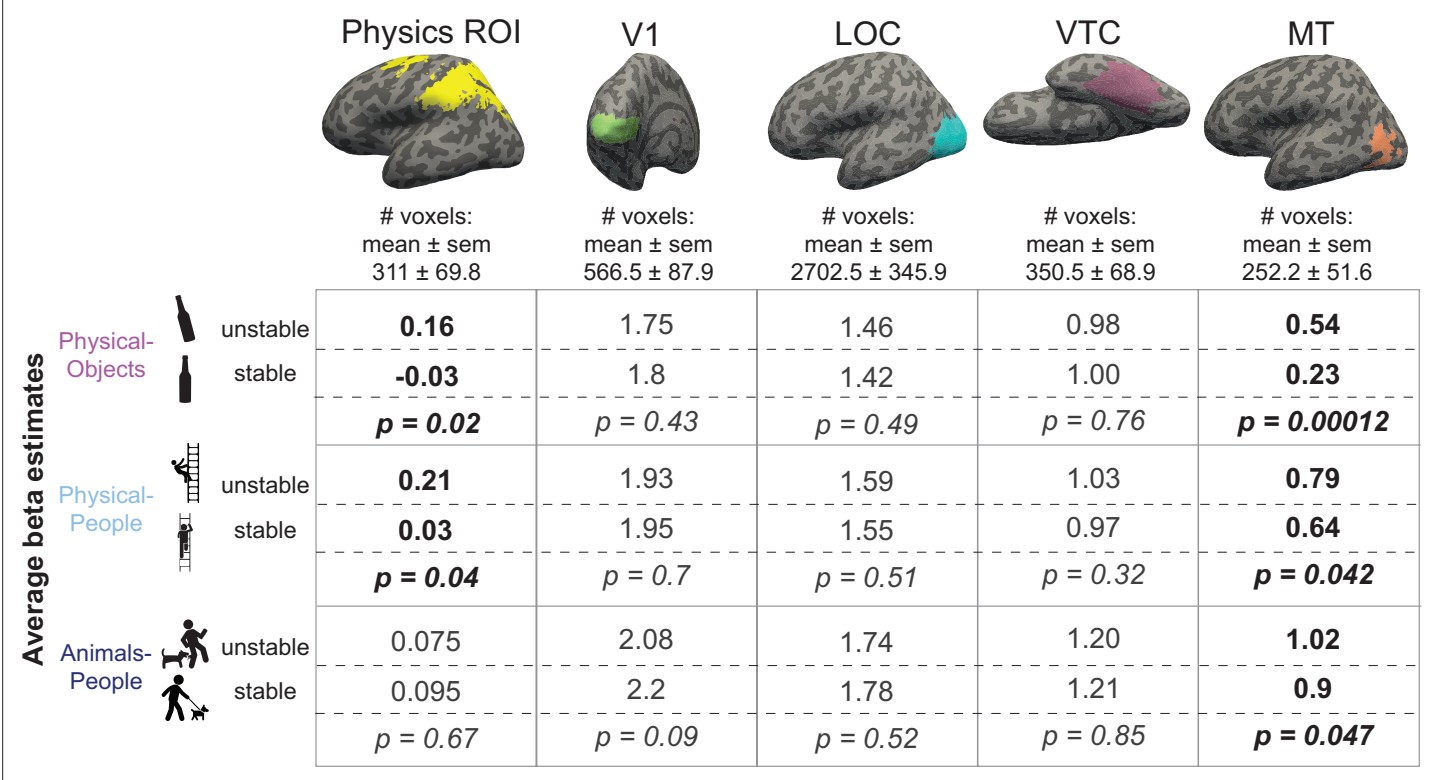

**Figure 5.** Average beta values for unstable and stable conditions in each of the scenarios. Columns represent different functional regions of interest (fROIs). Each cell shows average GLM estimated beta values for unstable and stable conditions along with the p value for a paired *t*-test comparing the two sets of values. Scenarios showing significantly higher response to unstable scenes compared to stable scenes are highlighted in bold in each column.

the pattern activations. This increased response to conditions with predicted motion is reminiscent of previous reports showing greater response in MT for static images with implied motion (***Kourtzi and Kanwisher, 2000***). The critical difference, however, is that whereas the earlier study reported higher responses in MT for static images depicting motion events happening at the moment the photograph was taken (implied motion) the current study shows activation of MT for motion that is only *predicted* (in the Physical-Objects and Physical-People scenarios). Most importantly, the lack of this univariate difference for the Animals-People scenario in the Physics ROI suggests that the Physics Network is not merely responding to general 'likeliness to change' or predicted motion.

## Discussion

Here, we report that frontoparietal cortical regions previously implicated in intuitive physical inference contain abstract information about physical stability, but representations in the ventral visual object recognition pathway, and those in feedforward CNNs trained on object recognition, do not. Our results indicate that representations in systems that are highly effective at invariant object recognition do not automatically support the general ability to distinguish physically stable from unstable scenes. Instead, this ability in humans is supported by a different system in the dorsal visual pathway that has been previously implicated in intuitive physical inference. This Physics Network (but not the ventral pathway) further shows a higher univariate response to unstable than stable scenes, as predicted if it performs forward simulations of what will happen next. Control analyses confirmed that neither pattern nor univariate information about physical stability in the Physics Network can be straight-forwardly accounted for by low-level visual features, differential eye movements or attention. Taken together, these results suggest that the human brain represents the physical world not via simple pattern classification but instead by building a model of the physical world that supports prediction via forward simulation.

Our study builds upon earlier work that has implicated regions in the parietal and frontal lobes in intuitive physical inference. Across multiple studies, similar frontoparietal regions have been shown to (1) respond more during physics-based tasks compared to color-based or social prediction tasks (*Fischer et al., 2016*), (2) represent physical concepts in verbal stimuli (*Mason and Just, 2016*), and (3) contain scenario-invariant representations of object mass (*Schwettmann et al., 2019*). Intriguingly, the regions activated during physical inference overlap with regions shown previously to be engaged during visually guided action (*Goodale and Westwood, 2004*) and tool use (*Gallivan et al., 2013*; *Valyear et al., 2007*; *Mruczek et al., 2013*), perhaps because these tasks also require a representation of the physical world. Indeed physical simulation has been proposed as a crucial component in models for human-like robotic action planning (*Toussaint et al., 2018*; *Toussaint et al., 2020*; *Mordatch et al., 2012*) and flexible tool use (*Allen et al., 2020*; *Osiurak and Badets, 2016*; *Osiurak et al., 2020*); plausibly the same brain mechanisms could have arisen or adapted to support all of these computations. Note however that these same regions also overlap with the 'multiple demand' system (*Fischer et al., 2016*), and are unlikely to be selectively engaged in *only* physical inference. In the present study, we strengthen evidence that these regions are engaged in intuitive physical inference by showing that they carry a new kind of physical information: the dynamic physical stability of objects and people in a scene. Interestingly, this information is present when participants simply view images of physical scenes, even though they are not asked to judge stability.

More importantly, our work speaks not only to which brain regions are implicated in intuitive physical inference, but what kinds of computations these inferences entail in both minds and machines. In particular, it is a matter of active current debate in AI whether features extracted by feedforward computations in networks trained on visual object classification will suffice for physical inference, or whether richer internal models that support forward simulation are required. Some work has shown that CNNs can learn from extensive labeled datasets to infer the stability of block towers (*Lerer et al., 2016*; *Conwell et al., 2019*) and predict future outcomes (*Lerer et al., 2016*; *Conwell et al., 2019*; *Li et al., 2016*). But these CNNs have so far been tested only on specific tasks and stimuli, and we show here that they do not generalize across scenarios for the task of determining physical stability. Instead, it has been proposed that a general ability to predict what will happen next in physical scenarios will require a more structured representation of the physical world that will support forward simulation (*Battaglia et al., 2013*; *Ullman et al., 2017*). A parallel debate is raging in cognitive science (*Lerer et al., 2016*; *Conwell et al., 2019*; *Battaglia et al., 2013*; *Firestone and Scholl, 2017*; *Firestone and Scholl, 2016*; *Ludwin-Peery et al., 2019*; *Davis and Marcus, 2015*; *Chater and Oaksford, 2017*; *Davis et al., 2017*; *Ludwin-Peery et al., 2020*), between those who argue that because human physical inferences occur rapidly (*Firestone and Scholl, 2017*) and preattentively (*Firestone and Scholl, 2017*) they are computed by something like a pattern recognition process, versus those who argue that human and primate physical inference behavior is best accounted for by mental simulation (*Battaglia et al., 2013*; *Ullman et al., 2017*; *Zhang et al., 2016*; *Gerstenberg et al., 2017*; *Rajalingham et al., 2021*). Three lines of evidence from the present study indicate that pattern recognition alone – as instantiated in feedforward CNNs and the ventral visual pathway – is unlikely to explain physical inference in humans, at least for the case of physical stability. First, we find generalizable representations of physical stability in the brain that we do not find in CNNs. Second, these abstract representations of stability are not found in the ventral visual pathway, which is thought to conduct pattern classification and is well modeled by CNNs, but rather in the dorsal pathway, previously implicated in intuitive physical inference. Third, we find a higher univariate response in this Physics Network for unstable scenes, where there is more to simulate, than the stable scenes, where nothing is predicted to happen next. However, note that our hypothesis pertains only to feedforward CNNs trained on object recognition (on ImageNet dataset); It remains an open question whether a feedforward CNN extensively trained on naturalistic images of stability may learn generalizable representations of physical stability.

We therefore hypothesize that visual information represented in the ventral visual cortex is used by the Physics Network for efficient inference of physical properties (see *Almeida et al., 2013*; *Kravitz et al., 2011* for the cases of tool processing and visuospatial reasoning), and this representation is in turn used for forward simulation of what will happen next. This idea has been recently proposed as an integrated computational model that uses visual representations from a deep-learning-based inverse graphics model to initialize simulations in a physics-engine-based generative model of object

dynamics, which then can be used to perceive, predict, reason about, and plan with physical objects (*Yildirim et al., 2019*; *Wu et al., 2015*; *Wu et al., 2017*). This class of models with flexible, object-centric representations and the ability to learn from realistic visual inputs (*Bear, 2020*; *Mrowca, 2018*) should be able to make predictions of physical stability on the realistic stimuli used in our experiment and also form the basis for neurally mappable encoding models of the candidate physics regions. The question of how such a model is instantiated, if at all, in the brain remains unanswered and provides a fertile avenue for future exploration.

Many questions remain. First, physical stability is just one of many aspects of physical scene understanding. Future investigations can explore whether the Physics Network also represents other physical properties of objects (like friction and elasticity), relational attributes (like support, containment, attachment), and physical forces and events. Second, if indeed the Physics Network is conducting forward simulations, when exactly does it run and how detailed are its simulations? According to one hypothesis, our mental physics engine compresses the rich details in our visual world into a relatively small number of individual objects and associated events in order to efficiently generate a reasonable approximation of the scene at the spatial and temporal scales relevant to human perception and action (*Ullman et al., 2017*). These abstractions may also enable us to run simulations faster than real time with compressed timescales (like hippocampal replay, *Lee and Wilson, 2002*), enabling us to make rapid and accurate predictions of the consequences of multiple actions under consideration, including our ability to make fast and automatic physical inference (*Firestone and Scholl, 2017*; *Firestone and Scholl, 2016*). Third, is the same neural machinery underlying simulation of the external physical world also recruited when we consider the consequences of our own actions? Answering this question would help elucidate how action planning and tool use are related to the neural system for physical inference, given that much of the Physics Network lies adjacent to or overlaps with brain regions engaged in action planning and tool use (*Gallivan et al., 2013*; *Valyear et al., 2007*; *Mruczek et al., 2013*; *Gallivan et al., 2011*; *Gallivan and Culham, 2015*).

## Materials and methods

### Participants

Thirteen subjects (ages 21–34; 6 females and 7 males) participated in the experiment which was cut short due to COVID-19 pandemic. All participants were right handed and had normal or corrected-to-normal vision. Before participating in the experiment, all subjects gave informed consent to the experimental protocol approved by the Massachusetts Institute of Technology (MIT) Committee on the Use of Humans as Experimental Subjects (#0403000096). The study was conducted in compliance with all the relevant ethical guidelines and regulations for work with human participants.

### Stimuli

All images were chosen to belong to six different experimental conditions divided into three scenarios with two conditions each (see *Figure 1B* for examples): objects in stable or unstable conditions ('Physical-Objects'); people in physically stable or unstable conditions ('Physical-People'); and, people with animals in perilous (unstable) or relatively safe (stable) conditions ('Animals-People'). The stimulus set included 14 images for each condition in each scenario. All images were scaled so that the longer dimension measured 8° on the projector screen places ~136 cm from the participant.

#### Screening using deep neural networks

We chose images from each of these six experimental conditions such that stability decoding accuracy was close to chance both within the scenario and also across scenarios on features extracted from the initial layers of a deep CNN. The main goal of this selection process was to screen images to minimize potential confounds of low-level visual features on stability decoding. First, we rescaled and padded each image with pixels of zero brightness to obtain images measuring 224 × 224 pixels. Then, we extracted features from the first pooling layer ('pool_1') of a feedforward convolutional network, VGG-16, trained on ImageNet object classification task. We then trained separate linear SVM classifiers with fourfold cross-validation for each scenario separately to distinguish between stable and unstable images. The classification accuracies were close to chance (=50%) for all three scenarios (% accuracy: 42.9%, 53.6%, and 42.9% for Physical-Objects, Physical-People,

and Animals-People scenarios, respectively). We then tested each classifier on the remaining two scenarios to quantify cross-decoding of stability across scenarios. Here also, we found close to chance (=50%) cross-decoding performance (average % accuracy = 51.8% for Physical-Objects vs. Physical-People, 44.6% for Physical-Objects vs. Animals-People, and 41.1% for Physical-People vs. Animals-People).

## Interestingness ratings

In order to minimize the influence of differential attention or interest on our results, we set out to quantify how interesting or attention-grabbing our stimuli are. We ran a behavioral experiment on 11 subjects (9 of them had previously participated in the fMRI part of this study) where we asked them to rate how *interesting* they found an image to be on a scale of 1–5 (1 – least interesting and 5 – most interesting). To validate this measure, we also included high and low arousal images ($n$ = 14 each) from IAPS (*Lang et al., 1997*) and obtained higher interestingness ratings for high arousal images compared to low arousal IAPS images (avg interestingness rating: high arousal images = 3.5 and low arousal images = 1.5; $p < 0.000005$). Further, as expected, subjects found unstable condition to be more interesting than stable condition across scenarios (*Supplementary file 1*, last column). However, we found that this difference did not significantly interact with scenario using an ANOVA ($p = 0.23$).

## Experimental design

### Physics ROI localizer

Each participant performed two runs of an 'intuitive physics' fMRI localizer task previously used to functionally define the frontoparietal physics engine in the brain (*Fischer et al., 2016*; *Schwettmann et al., 2019*). In this task, subjects viewed short movies (~6 s) depicting unstable towers made of blue, yellow and white blocks (see *Figure 1A*) created using Blender 2.70 (Blender Foundation). The tower was centered on a floor that was colored green on one half and red on the other half such that it would topple toward one of the halves if gravity were to take effect. Throughout the movie, the tower remained stationary while the camera panned 360° to reveal different views of the tower. Subjects viewed these movies and were instructed to report whether more blocks would come to rest on the red or green half of the floor ('physics' task), or whether there are more blue or yellow blocks in the tower ('color' task).

Each run of this localizer task consisted of 23 18 s blocks: 3 fixation-only blocks, 10 blocks each of the physics and the color task. Each 6 s movie was preceded by a text instruction displayed on the screen for 1 s which read either 'where will it fall?' ('physics' task) or 'more blue or yellow?' ('color' task) and was followed by a 2-s response period with a blank screen. This sequence was repeated twice within a block with the same task cue but different movies. The subjects responded by pressing one of two buttons on a response box for each alternative in a task. The mapping of the buttons to the response was switched for the second run to rule out the effects of specific motor responses on the observed neural activations. We used a *physics task > color task* contrast to functionally identify the frontoparietal physics regions in each subject individually.

### Stability experiment

In addition to the physics ROI localizer, each participant also performed four runs of the main experiment. In this experiment, subjects viewed a sequence of images while maintaining fixation on a red dot at the center of the image and performed a 1-back task.

Each run of this experiment contained 3 rest/fixation blocks and 12 20-s stimulus blocks (2 blocks for each of the 6 experimental conditions). Each run began with a fixation-only block followed by a random ordering of the blocks corresponding to the six experimental conditions. This was followed by another fixation-only block and the six experimental condition blocks shown in the reverse order. Each run ended with another fixation-only block. Each image block contained 10 trials including 2 1-back trials. In each trial, an image was shown for 1.8 s followed by 0.2 s of fixation-only interval. Subjects were instructed to maintain fixation (confirmed using eye-tracker for 6 out of the 13 subjects) and respond by pressing a button on the response box whenever the same image repeated one after the other in the sequence.

## CNN analysis

Activations from the fc1000 layer (fully connected layer just preceding the softmax layer) of a Resnet-50 model trained on ImageNet object recognition challenge were extracted for both stable and unstable conditions across Block Towers, Physical-Objects, and Physical-People scenarios. The Physical-Objects and Physical-People scenarios are exactly the same as in the fMRI experiment, and the Block Towers set ($n$ = 258 each for stable and unstable conditions) was used to replicate and extend earlier findings. A linear Support Vector Classifier (SVM) was trained to distinguish between stable and unstable conditions within each scenario using fourfold cross-validation. To test the generalizability of the learned classifier to other scenarios, the SVM classifier was tested on stability detection in the remaining two scenarios. We replicated the results using fc8 layer activations in an ImageNet pretrained VGG-16 network. The statistical significance of classification accuracy was estimated by comparing it to a null distribution of accuracies computed by randomly shuffling the training labels 10,000 times and measuring the test accuracy each time.

## Data acquisition

All imaging was performed on a Siemens 3T MAGNETOM Tim Trio scanner with a 32-channel head coil at the Athinoula A. Martinos Imaging Center at MIT. For each subject, a high-resolution T1-weighted anatomical image (MPRAGE: TR = 2.53 s; TE = 1.64, 3.44, 5.24, 7.04 ms; $\alpha$ = 7°; FOV = 220 mm; matrix = 220 × 220; slice thickness = 1 mm; 176 slices; acceleration factor = 3; 32 reference lines) was collected in addition to whole-brain functional data using a T2*-weighted echo planar imaging pulse sequence (TR = 2 s; TE = 30 ms; $\alpha$ = 90°; FOV = 216 mm; matrix = 108 × 108; slice thickness = 2 mm; voxel size = 2 × 2 mm in-plane; slice gap = 0 mm; 69 slices).

## Eye movement recordings

Eye movement data were recorded from 6 of the 13 subjects during both the physics ROI localizer task and the stability experiment using the EyeLink 1,000 Eye-Tracker (SR Research) inside the scanner. We could not collect eye movement data from other subjects due to technical difficulties. Eye tracking data were preprocessed and analyzed to confirm that eye movements could not explain differences in BOLD activity for various experimental conditions. For each trial in both the localizer and stability tasks, we computed the average abscissa and ordinate of the eye position, the number of saccades, average duration, and amplitude of saccades for the duration of the trial. We then performed $t$-tests to compare the average values of the aforementioned eye movement variables for stable and unstable conditions in each scenario across subjects.

## fMRI data preprocessing

Preprocessing was done using FreeSurfer (https://freesurfer.net/). All other analyses were performed in MATLAB 2015B (The Mathworks). fMRI data preprocessing included motion correction, slice time correction, linear fit to detrend the time series, and spatial smoothing with a Gaussian kernel (FWHM = 5 mm). Before smoothing the functional data, all functional runs were coregistered to the subject's T1-weighted anatomical image. All analyses were performed in each subject's native volume and in some cases the results were plotted on the subject's native inflated cortical surface only for better visualization (using FreeSurfer's mri_vol2surf function). The general linear model included the experimental conditions and 6 nuisance regressors based on the motion estimates ($x$, $y$, and $z$ translation; roll, pitch, and yaw of rotation).

## Group-level physics parcel

We derived group-level physics parcels from the localizer data (*Schwettmann et al., 2019*) in 27 subjects using the Group-constrained Subject-Specific method described previously (*Julian et al., 2012*). Briefly, individual subjects' binary activation maps (p < 0.005 uncorrected) were overlaid on top of each other in MNI space. This overlap map was spatially smoothed with a Gaussian filter (FWHM = 8 mm) and then thresholded so that the map contained only those voxels with at least 10% overlap across subjects. Then, the overlap map was divided into group-level parcels using a watershed image segmentation algorithm (*watershed* function in MATLAB). Finally, we selected a subset of parcels in which at least 16 out of the 27 (~60%) subjects show some activated voxels. This resulted in seven group-level parcels spanning frontal, parietal, and occipital lobes. We rejected two of the seven

parcels in the occipital cortex since they were shown to respond to both physical and social stimuli (**Fischer et al., 2016**). The remaining parcels correspond coarsely to previously described physics regions (**Fischer et al., 2016**; **Schwettmann et al., 2019**), however, we believe that the new parcels are probably more stable because they are derived from a larger subject pool. These five parcels were then combined to get one group-level parcel each in left and right hemispheres. We will make the parcels publicly available.

## fROI definition

We defined fROI in each individual subject as the intersection of subject-specific localizer contrast map and group-level (or anatomical) parcels. Specifically, we used the physics localizer to identify brain regions in each individual subject that responded more to the physics task compared to the color task (uncorrected p value <0.001 for the physics > color contrast). This contrast map was then intersected with the group-level physics parcels created from the physics localizer data collected in a previous study (**Schwettmann et al., 2019**). Thus, the individual subject fROI contained only those voxels that showed significantly stronger activations for the physics task compared to the color task and fell within the group-level physics parcel. This allowed the fROI locations and sizes to vary across subjects but restricted them to a common general region across subjects. In addition to the physics fROI, we also defined fROIs for the primary visual cortex (V1), LOC, and VTC in each subject. As before, we used data from the physics localizer to identify brain regions that responded to visual stimuli compared to fixation (uncorrected p value <0.001 for physics + color > fixation contrast). We then intersected this contrast map with masks derived from anatomical parcellation and considered only those significant voxels from the contrast map lying within the anatomical mask for further analyses. In addition, we also localized area MT using a standard localizer (coherent vs. random dot motion contrast) (**Tootell et al., 1995**) and included significant voxels within an anatomically defined mask (taken from Destrieux atlas in FreeSurfer).

## Multivoxel pattern correlation analysis

### Within category separability

To assess if the candidate physics fROI holds information about physical stability, we used the multivoxel pattern correlation analysis (**Haxby et al., 2001**). In an fROI, we computed *within* condition pattern correlations (correlation between voxel activation patterns for even and odd runs, computed separately for unstable and stable conditions within a scenario) and *between* conditions pattern correlations (correlation between voxel activation patterns for even runs of unstable condition and odd runs of stable condition within a scenario, and vice versa). We computed the *within* and *between* condition pattern correlations for each scenario in each hemisphere in each subject and compared the magnitudes of correlations using appropriate statistical tests after transforming the correlations using Fisher transform (*atanh* function in MATLAB). A significantly higher *within* condition pattern correlation compared to *between* condition pattern correlation indicates that unstable and stable conditions evoke distinctive voxel activation patterns in a given fROI. Note that by always comparing data across even and odd runs we avoid obscuring pattern information from temporal pattern drifts (**Alink et al., 2015**).

### Across category similarity

To explore the generalizability of neural representation of physical stability in a given fROI, we computed multivoxel pattern correlations computed across scenarios. Specifically, in a given fROI, we extracted activation patterns for unstable and stable conditions from the two scenarios under consideration (say, 'Physical-Objects' and 'Physical-People' scenarios). We then computed pattern correlations between unstable (or stable) conditions across scenarios (*within* condition), and pattern correlations between unstable and stable conditions across scenarios (*between* condition). In this way, we computed the four pairwise pattern correlations for each pair of categories within a given fROI in each hemisphere of each individual subject and transformed them using Fisher transformation. We compared the magnitudes of *within* and *between* condition correlations for each pair of categories across subjects using a paired *t*-test.

## Acknowledgements

This work was supported by NIH grant DP1HD091947 to N.K., a US/UK ONR MURI project (Understanding Scenes and Events through Joint Parsing, Cognitive Reasoning and Lifelong Learning), and National Science Foundation Science and Technology Center for Brains, Minds, and Machines Grant CCF-1231216. The Athinoula A Martinos Imaging Center at MIT is supported by the NIH Shared instrumentation grant #S10OD021569. We thank Kirsten Lydic for helping with data collection, David Beeler for helping with fMRI data preprocessing, and Ilker Yildirim, Kevin Smith, and members of the Kanwisher Lab for helpful discussions.

## Additional information

### Funding

| Funder | Grant reference number | Author |
| --- | --- | --- |
| National Institutes of Health | DP1HD091947 | Nancy Kanwisher |
| Office of Naval Research | MURI | Joshua B Tenenbaum<br>Nancy Kanwisher |
| National Science Foundation | CCF-1231216 | Joshua B Tenenbaum<br>Nancy Kanwisher |

The funders had no role in study design, data collection, and interpretation, or the decision to submit the work for publication.

### Author contributions

RT Pramod, Conceptualization, Data curation, Formal analysis, Writing – original draft, Writing – review and editing; Michael A Cohen, Conceptualization, Writing – review and editing; Joshua B Tenenbaum, Conceptualization, Funding acquisition, Supervision, Writing – review and editing; Nancy Kanwisher, Conceptualization, Funding acquisition, Resources, Supervision, Writing – review and editing

### Author ORCIDs

RT Pramod http://orcid.org/0000-0002-5933-7893
Nancy Kanwisher http://orcid.org/0000-0003-3853-7885

### Ethics

All subjects gave informed consent to the experimental protocol approved by the Massachusetts Institute of Technology (MIT) Committee on the Use of Humans as Experimental Subjects (#0403000096). The study was conducted in compliance with all the relevant ethical guidelines and regulations for work with human participants.

### Decision letter and Author response

Decision letter https://doi.org/10.7554/eLife.71736.sa1
Author response https://doi.org/10.7554/eLife.71736.sa2

## Additional files

### Supplementary files

• Supplementary file 1. Average values of eye tracking variables and interestingness ratings for stable and unstable conditions in all three scenarios. Each cell shows the average value of the variable for stable and unstable conditions along with the p value for a paired *t*-test comparing the two sets of values across subjects. The first five columns correspond to eye movement variables collected on 6 subjects during the fMRI experiment and the last column is for the interestingness rating collected on the same set of images but in 11 subjects outside the scanner (see Methods for details). Since the subjects were instructed to maintain fixation at the center of the image, we did not observe any saccadic events (amplitude >1°) and hence we are calling the small ballistic events as simply *eye movements*. Significant effects (p < 0.05) are highlighted in bold.

• Supplementary file 2. MVPA in the candidate physics regions computed separately for parietal and frontal ROIs. Each cell shows the average Fisher transformed within and between stability condition

pattern correlations along with the p value for a paired *t*-test comparing the two sets of values. Each column includes the results from one fROI. The top three rows contain results for within-scenario pattern correlation analysis and the bottom three rows show results for the pattern correlation analysis across scenarios. Significant effects with within condition correlations greater than between condition correlations are highlighted in bold.

• Supplementary file 3. Average beta values for unstable and stable conditions in each of the scenarios computed separately for parietal and frontal physics ROIs. Each cell shows average GLM estimated beta values for unstable and stable conditions along with the p value for a paired *t*-test comparing the two sets of values across subjects. Scenarios showing significantly higher response to unstable scenes compared to stable scenes are highlighted in bold in each column.

• Supplementary file 4. Reaction time analysis for stable and unstable conditions in each of the scenarios for the stability judgment task.

• Supplementary file 5. Accuracies for stable and unstable conditions in each of the scenarios for the stability judgment task.

• Supplementary file 6. Methods and results for the stability detection behavioral task.

• Transparent reporting form

## Data availability

All data generated and analyzed during this study will be made available at https://osf.io/xc3t8/.

The following dataset was generated:

| Author(s) | Year | Dataset title | Dataset URL | Database and Identifier |
|---|---|---|---|---|
| Cohen M, Tenenbaum J, Kanwisher N | 2022 | Data from: Invariant Representation of Physical Stability in the Human Brain | https://osf.io/xc3t8 | OSF, 10.17605/OSF.IO/XC3T8 |

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
