## [Editor Report]

This is an intriguing study using cleverly designed stimuli to investigate the representation of physical stability in the human brain. This paper will be of interest to readers wondering when human cognition uses pattern matching similar to that used by machine learning algorithms, and when it relies on more specialised processes evolved for specific tasks. The well-crafted experiments provide convincing support for the authors' claim that the human brain computes an abstract representation of the physical stability of visual scenes.

---

## [Decision Letter]

**Decision letter after peer review:**

Thank you for submitting your article "Invariant representation of physical stability in the human brain" for consideration by *eLife*. Your article has been reviewed by 3 peer reviewers, including Peter Kok as Reviewing Editor and Reviewer #1, and the evaluation has been overseen by Floris de Lange as the Senior Editor. The following individual involved in review of your submission has agreed to reveal their identity: Jaqueline Snow (Reviewer #3).

The reviewers have discussed their reviews with one another, and agreed that the study is of interest, but they had serious concerns about the interpretation of the main results. Several alternative explanations of the results will need to be ruled out conclusively, likely requiring new (behavioural) data, in order for the paper to be suitable for publication in *eLife*. Note that given the seriousness of the concerns, simply acknowledging them will not be sufficient. The reviewing editor has drafted this letter to help you prepare a revised submission, should you feel that you are able to resolve these issues.

Essential revisions:

1. One major concern with the interpretation of the findings is that while the stimuli in the Animals-People condition seem to be relatively straightforward to understand whether they are "unstable" or "stable" scenes, the stimuli in the Physical-People condition seem more difficult to comprehend when they are "unstable" versus "stable", perhaps due to their unusual scene composition. For instance, it can take a moment to understand what is going on in the scene depicting a man precariously perched atop a ladder, with the ladder balanced on one leg over a stairwell (Figure 2A). This leaves open the possibility that the elevated activity observed in the fronto-parietal areas may not reflect forward simulation of unstable (over stable) scenes involving physical objects and people, but instead, the requirements of parsing difficult-to-comprehend scenes (which happen to be unstable and physical) over straightforward scenes (which are either stable or involve animals). Put simply, the fMRI responses recorded in fronto-parietal cortex may have more to do with scene comprehension than with physical stability. This confound will need to be ruled out conclusively. This will likely require collecting and analysing new data. For instance, the authors could collect behavioral questionnaire data to gauge for each stimulus ease of comprehension, as well as anticipated motion and likelihood of change (see point 2). The behavioral data could then be correlated with the fMRI responses, and it can be tested whether fMRI results are explained better by physical stability or one or more of these other factors. Of course, the authors may have a different approach in mind, which is perfectly fine, as long as the confounds are resolved convincingly.

2. Can the effects of physical stability reported here be dissociated from effects of implied motion, or likeliness-to-change? In the current study, a higher response to unstable vs stable scenes in reported in motion area MT, which the authors describe in terms of implied motion. It therefore seems possible to describe the results in the "Physics Network" in terms of implied motion, rather than physical stability, as well. As for the first point, this alternative explanation will need to be ruled out conclusively.

3. The perilous-vs-non-perilous (Animals-People) condition is a crucial control for attention. However, it seems the relevant interaction, i.e. more generalization for stable-to-stable than stable-to-perilous, is not statistically tested. This test seems crucial to demonstrate that generalization is specific to physical stability.

4. The current study, and its findings, should be situated more deeply in the context of what has been done before. Fischer et al., (2016) showed that specific regions of frontal and parietal cortex responded more strongly when observers performed a physical reasoning task about which direction a tower of blocks would fall (versus a control task). Schwettmann et al., (2019) examined how object mass is represented in these same dorsal areas, and in the ventral visual pathway. The differential pattern of results, including the finding of generalizability across scenarios, was the same as with the present findings, but for object mass, rather than scene stability. Yet Schwettmann et al., (2019) is only briefly mentioned despite its many methodological and conceptual similarities to the present study. The authors should more effectively situate the present findings relative to their previous work (in the Introduction), as well as explain what is novel and surprising about these most recent findings (in the Discussion).

5. Can the authors clarify why they used block towers, Physical-People scenes, and Physical-Objects scenes with the CNN, but used Physical-People scenes, Physical-Objects scenes, and Animals-People scenes with the human observers? Because the comparison of the CNN to human observers is central to the paper, it is worth explaining why different stimulus sets were used between the two.

6. It seems that the CNNs simply didn’t do a great job classifying physical stability in the first place (60% classification accuracy). In that case, how generalisable should we expect their representations to be anyway? It could be that a CNN that does achieve high accuracy on physical stability judgments (90%?) would actually show this kind of general transfer; but we don’t know that from the data presented here, because it’s possible that the lack of generality arises from poor performance to begin with.

7. The Animals-People results in Table 1, third row, seem quite puzzling. Is it right that there is a far higher between- than within-correlation in all ROIs, especially the visual ones? How can this be explained?

8. Overall, the generalization effects reported in Table 1 seem quite subtle, and there are of course many statistical tests reported in this Table; have these been corrected for multiple comparisons?

9. Could the authors explain why they chose to split even and odd runs in the within-scenario analysis but not in the between-scenario analysis. As acknowledged in the caption for Figure 2, this prevents direct comparison of the correlation values between the two analyses, so the justification for the difference should be made explicit.

10. The brief discussion of the generalization effect (Animals-People to Physical-People) in LOC is not very satisfactory. For instance, what kind of similar shape information could be driving this effect that is not present in the other generalization effects? Also, from Table 1 there seems to be a significantly negative generalization effect in V1 for Physical-Objects to Animals-People, but this is not discussed at all.

11. “We reasoned that if the candidate physics regions are engaged automatically in simulating what will happen next, they should show a higher mean response when viewing physically unstable scenes (because there is more to simulate) than stable scenes (where nothing is predicted to happen).” It seems true enough that, once one knows that a scene is stable, one doesn’t then need a dynamically updated representation of its unfolding. But the question that this paper is about is how we determine, in the first place, that a scene is stable or not. The simulations at issue are simulations one runs before one knows their outcome. Stable scenes may well have a lot to simulate, even if we determine after those hefty simulations that the scene is stable after all. And of course unstable scenes might well have very little to simulate, if the scene is simple and the instability is straightforwardly evident. Can the authors say more about why it’s easier to determine that a stable scene is stable than that an unstable scene is unstable?

12. “Interestingness” ratings seem like a not-quite-adequate approach for evaluating how attention-grabbing the towers were. Why not a measure like how much they distract from another task? The authors could add a new measure of attention-grabbing-ness with a sounder basis in the visual psychophysics literature, or a stronger justification for why “interestingness” ratings are up for the job.

13. More examples of the stimuli used in each scenario should be provided, either as a supplemental figure or in a hyperlink, to help readers evaluate the stimuli.

[Editors’ note: further revisions were suggested prior to acceptance, as described below.]

Thank you for resubmitting your work entitled “Invariant representation of physical stability in the human brain” for further consideration by *eLife*. Your revised article has been evaluated by Floris de Lange (Senior Editor) and a Reviewing Editor.

The reviewers agree that the manuscript has been significantly improved. There are a couple of remaining issues to be addressed before the paper can be accepted, as outlined below:

1. The response to previous point 6, regarding the generalization of the CNN classification, is not completely satisfactory. It’s still a worry that the paper is conflating (1) whether CNNs can learn to discriminate stable from unstable with (2) whether whatever it is CNNs do learn is generalizable. The issue raised earlier was that CNN performance was so low that, for all we know, better performance would have revealed that the CNNs’ generalizability wasn’t so bad after all. To say that the poor performance “reinforces our point that pattern recognition systems are not very good at discerning physical stability” still feels a bit off-target. There are multiple ways to not be good at something, and the authors here are claiming that one of those ways is a lack of generalizability per se. But our previous point was that we can’t tell the difference between bad generalizability and bad overall performance. If the authors themselves are making this distinction (and they are in the paper), it seems crucial to separate those claims and support them independently. If they cannot be supported directly from the data, it would be appropriate to discuss this as a limitation of the study.

2. The response to previous point 3, regarding the specificity of decoding of representations in the Physics network, did not fully address the issue raised. The authors infer specificity from a null effect in ‘decoding’ stability/peril in the Animal-People condition, but of course a null effect is not evidence for absence. It would be more appropriate to test whether the decoding in the Physical-Object and Physical-People conditions is statistically better than in the Animal-People condition. E.g. in an ANOVA with two factors: Physical stability vs Animate stability and Within vs. Between. In other words, in their reply the authors cited two findings, (a) a null effect in the Animals-People condition and (b) a significant effect in the Physics condition, but to support their claims it would be useful to test whether b is significantly larger than a. Otherwise, the findings could for instance be due to the Animals-People null effect being based on very noisy measurements, i.e. a non-informative null effect, which does not constitute evidence for the specificity of the physical stability effects.

---

## [Author Response]

Essential revisions:1. One major concern with the interpretation of the findings is that while the stimuli in the Animals-People condition seem to be relatively straightforward to understand whether they are “unstable” or “stable” scenes, the stimuli in the Physical-People condition seem more difficult to comprehend when they are “unstable” versus “stable”, perhaps due to their unusual scene composition. For instance, it can take a moment to understand what is going on in the scene depicting a man precariously perched atop a ladder, with the ladder balanced on one leg over a stairwell (Figure 2A). This leaves open the possibility that the elevated activity observed in the fronto-parietal areas may not reflect forward simulation of unstable (over stable) scenes involving physical objects and people, but instead, the requirements of parsing difficult-to-comprehend scenes (which happen to be unstable and physical) over straightforward scenes (which are either stable or involve animals). Put simply, the fMRI responses recorded in fronto-parietal cortex may have more to do with scene comprehension than with physical stability. This confound will need to be ruled out conclusively. This will likely require collecting and analysing new data. For instance, the authors could collect behavioral questionnaire data to gauge for each stimulus ease of comprehension, as well as anticipated motion and likelihood of change (see point 2). The behavioral data could then be correlated with the fMRI responses, and it can be tested whether fMRI results are explained better by physical stability or one or more of these other factors. Of course, the authors may have a different approach in mind, which is perfectly fine, as long as the confounds are resolved convincingly.

Thank you for raising this important point. To test whether images in the three scenarios systematically differed in ease of understanding, we ran a behavioral task where we asked subjects (n = 16) to categorize the stability of each image (unstable/peril vs stable/non-peril). We found that reaction times differed significantly between stable and unstable conditions within each scenario, but the direction of the effect was not consistent across the scenarios (see Author response table 1) and – most importantly – the overall pattern of RT and accuracy is not consistent with an account of our fMRI findings in terms of difficulty. Specifically, although the reviewer is correct that the Animals-People condition was easier overall than the Physical-People condition both in terms of accuracy (p = 0.047 for signrank test on avg accuracy across subjects) and reaction time (p = 0.004 for signrank test on avg RT across subjects), it is the difference in difficulty between the unstable/perilous conditions versus the stable/nonperilous conditions that is important because it is this comparison that our fMRI results are based on. And here, we find that in fact subjects were *faster* (p = 0.002) and *more accurate* (p = 0.08; tending towards significance) on the unstable than stable conditions in the Physical-People scenarios, an effect that goes in the opposite direction from the difficulty hypothesis. Further, although participants did take more time to categorize the unstable than stable stimuli in the Physical-Objects condition (p = 0.05), this effect was smaller in magnitude than that found for the Animals-People control scenario (p = 0.033 for interaction between scenario and stability in an ANOVA) where we did not find similarly higher activations for unstable/perilous than stable/nonperilous conditions. Thus, our behavioral data are not consistent with the hypothesis that any of our important fMRI findings (either univariate or multivariate) reflect difficulty confounds.

**Author response table 1. sa2table1:** 

	Physical-Objects	Physical-People	Animals-People
Stable	Unstable	Stable	Unstable	Non-Peril	Peril
% Accuracy (mean ± std across subjects)	94.4 ± 1.9	93.1 ± 1.9	88.5 ± 3.3	95.7 ± 1.4	93.5 ± 1.4	99.5 ± 0.4
p-value (signrank test on avg. accuracy across subjects)	p = 0.67	p = 0.008	p = 0.002

**Author response table 2. sa2table2:** 

	Physical-Objects	Physical-People	Animals-People
Stable	Unstable	Stable	Unstable	Non-Peril	Peril
Reaction Time, ms(mean ± std)	862.1 ± 43.8	943.2 ± 45.2	1088.2 ± 48.8	958.1 ± 46.8	1003.2 ± 60.8	881.5 ± 29.9
p-value(signrank test on avg. RTs across subjects)	p = 0.05	p = 0.002	p = 0.008
Average RT, ms	902.7	1023.2	942.3
RT difference, ms(unstable – stable)	81.2	-130.1	-121.7

For a discussion of anticipated motion and likelihood of change as potential confounds, please see responses to point 2 below.

2. Can the effects of physical stability reported here be dissociated from effects of implied motion, or likeliness-to-change? In the current study, a higher response to unstable vs stable scenes in reported in motion area MT, which the authors describe in terms of implied motion. It therefore seems possible to describe the results in the “Physics Network” in terms of implied motion, rather than physical stability, as well. As for the first point, this alternative explanation will need to be ruled out conclusively.

We do indeed show higher responses to unstable (or perilous) scenes compared to stable (or non-perilous) scenes in area MT for all three scenario types, which we interpret as probably reflecting the encoding of ‘implied’ or ‘predicted’ motion. Importantly, however, processing of implied or predicted motion cannot explain results in the Physics Network because our key control condition (Animals-People) does *not* show higher activations for unstable/perilous compared to stable/nonperilous stimuli. Thus, although the unstable/perilous conditions in all three scenario types activate MT, presumably reflecting implied motion, the Animals-People scenario does *not* activate the “Physics Network”, and hence the activation of the Physics Network cannot be explained by implied motion alone.

3. The perilous-vs-non-perilous (Animals-People) condition is a crucial control for attention. However, it seems the relevant interaction, i.e. more generalization for stable-to-stable than stable-to-perilous, is not statistically tested. This test seems crucial to demonstrate that generalization is specific to physical stability.

We did report the stats requested here in Figure 2:

a) The two rightmost bars in Figure 2B show that we cannot decode stability/peril from the Animals-People conditions: the correlation of patterns within stability is no higher than the correlation of patterns between stability.

b) Further, the between-scenario comparisons in Figure 2D show that within-stability correlations are significantly greater than between-stability correlations only when generalizing across the Physical-Object and Physical-People scenarios, not when generalizing between either of the Physical conditions and the Animal condition, exactly as predicted by our hypothesis.

These analyses show that representations in the “Physics network” concern physical stability, not animate stability/peril.

If there is another analysis the reviewer would like to see let us know.

4. The current study, and its findings, should be situated more deeply in the context of what has been done before. Fischer et al., (2016) showed that specific regions of frontal and parietal cortex responded more strongly when observers performed a physical reasoning task about which direction a tower of blocks would fall (versus a control task). Schwettmann et al., (2019) examined how object mass is represented in these same dorsal areas, and in the ventral visual pathway. The differential pattern of results, including the finding of generalizability across scenarios, was the same as with the present findings, but for object mass, rather than scene stability. Yet Schwettmann et al., (2019) is only briefly mentioned despite its many methodological and conceptual similarities to the present study. The authors should more effectively situate the present findings relative to their previous work (in the Introduction), as well as explain what is novel and surprising about these most recent findings (in the Discussion).

There are many contexts that the present study can be situated in. Certainly, the prior work from our lab by Fischer et al., (2016) and Schwettmann et al., (2019) is highly relevant and these studies are accordingly cited in the introduction, and in detail in the Discussion. We could have framed this study as primarily a question about the “Physics Network” that was investigated in those studies, in which case they would have figured more prominently in the introduction. However, we felt the broader and more interesting theoretical framing of this work was not simply to functionally characterize a region of the brain, but to ask more fundamentally about the computations that in principle could, and that in practice *actually* do, enable humans to reason about the physical world. To this end we start our introduction with the importance to human survival of discerning whether the current situation is stable and hence likely to remain the same, or unstable and hence likely to change in the near future, and the question of whether this information can be extracted via pattern classification systems or requires simulations in a model of the world. We then conduct a preliminary test of that idea using CNN modeling, which leads to our predictions about where we will find this information in the brain. We think this framework is deeper and more interesting than beginning with an extensive review of our own prior fMRI studies.

We do however prominently discuss the relevant prior fMRI work in detail in the second paragraph of the discussion:

“Our study builds upon earlier work that has implicated regions in the parietal and frontal lobes in intuitive physical inference. Across multiple studies, similar fronto-parietal regions have been shown to (a) respond more during physics-based tasks compared to color-based or social prediction tasks^16^, (b) represent physical concepts in verbal stimul^23^, and (c) contain scenario-invariant representations of object mass^17^. Intriguingly, the regions activated during physical inference overlap with regions shown previously to be engaged during visually-guided action^24^ and tool use^25–27^, perhaps because these tasks also require a representation of the physical world. Indeed physical simulation has been proposed as a crucial component in models for human-like robotic action planning^28–30^ and flexible tool use^31–33^; plausibly the same brain mechanisms could have arisen or adapted to support all of these computations. Note however that these same regions also overlap with the “multiple demand” system^16^, and are unlikely to be selectively engaged in only physical inference.”

And we then go on to say exactly what our study contributes, that goes beyond that earlier work:

“In the present study, we strengthen evidence that these regions are engaged in intuitive physical inference by showing that they carry a new kind of physical information: the dynamic physical stability of objects and people in a scene… our work speaks not only to which brain regions are implicated in intuitive physical inference, but what kinds of computations these inferences entail in both minds and machines. In particular, it is a matter of active current debate in AI whether features extracted by feedforward computations in networks trained on visual object classification will suffice for physical inference, or whether richer internal models that support forward simulation are required. … Three lines of evidence from the present study indicate that pattern recognition alone – as instantiated in feedforward CNNs and the ventral visual pathway – is unlikely to explain physical inference in humans, at least for the case of physical stability. First, we find generalizable representations of physical stability in the brain that we do not find in CNNs. Second, these abstract representations of stability are not found in the ventral visual pathway, which is thought to conduct pattern classification and is well modeled by CNNs, but rather in the dorsal pathway, previously implicated in intuitive physical inference. Third, we find a higher univariate response in this Physics Network for unstable scenes, where there is more to simulate, than the stable scenes, where nothing is predicted to happen next.”

We think this is the strongest framing for our work, that it does justice to the prior fMRI work, and that it explains how our findings go beyond the earlier work. However, we welcome suggestions from the reviewer if they think we have left out important information.

5. Can the authors clarify why they used block towers, Physical-People scenes, and Physical-Objects scenes with the CNN, but used Physical-People scenes, Physical-Objects scenes, and Animals-People scenes with the human observers? Because the comparison of the CNN to human observers is central to the paper, it is worth explaining why different stimulus sets were used between the two.

Block towers have been used in previous computational modeling studies, and hence we also included them in our CNN studies here. However stable and unstable block towers usually differ in their low-level visual properties (stable towers are short and wide whereas unstable towers are narrow and tall, and they can be decoded from early layers of a pre-trained CNN), which is a confound we wanted to avoid in the fMRI experiment.

6. It seems that the CNNs simply didn't do a great job classifying physical stability in the first place (60% classification accuracy). In that case, how generalisable should we expect their representations to be anyway? It could be that a CNN that does achieve high accuracy on physical stability judgments (90%?) would actually show this kind of general transfer; but we don't know that from the data presented here, because it's possible that the lack of generality arises from poor performance to begin with.

You are correct in noting that CNNs don’t do a great job in classifying physical stability, which reinforces our point that pattern recognition systems are not very good at discerning physical stability. In fact, the classification accuracy that we have reported is close to the baseline performance in literature (Lerer et al., 2016). Interestingly, training on the block tower dataset itself could only bring up the stability classification accuracy to 68.8% on the real-world block tower images. While this is true of the current best model of stability detection, we think that CNNs trained on large-scale datasets of stability under varying scenarios may in future be able to potentially generalize to other natural scenarios. However, to our knowledge no such datasets exist.

7. The Animals-People results in Table 1, third row, seem quite puzzling. Is it right that there is a far higher between- than within-correlation in all ROIs, especially the visual ones? How can this be explained?

Thank you for noticing this issue! We should have noticed it ourselves, since the much higher correlations for between than within made no sense. Upon further checking the analysis code, we discovered that only the variable for within condition correlations in the Animals-People scenario was not being updated after computing the fisher z-transform. We have now fixed this issue and updated the corresponding table and figure, and the nonsensical finding is gone.

8. Overall, the generalization effects reported in Table 1 seem quite subtle, and there are of course many statistical tests reported in this Table; have these been corrected for multiple comparisons?

The generalization effects are indeed quite subtle in effect size, as is typical for MVPA analyses, but they are statistically robust. Correction for multiple comparisons would be required if we were performing multiple statistical tests for a hypothesis and a significant effect on *any one* of those tests would be considered as evidence for the hypothesis. However, in our case, our primary hypothesis is concerned with the Physics ROI and this hypothesis is accepted only if *all* six statistical tests came out as they did. That is, our hypothesis would not be accepted if only one of those tests came out significant. Thus, correcting for multiple comparisons is not required in this case. Moreover, the outcome of statistical tests in other ROIs have no bearing on our primary hypothesis about the Physics ROI, so they do not count as multiple comparisons for the tests performed on the Physics ROI.

9. Could the authors explain why they chose to split even and odd runs in the within-scenario analysis but not in the between-scenario analysis. As acknowledged in the caption for Figure 2, this prevents direct comparison of the correlation values between the two analyses, so the justification for the difference should be made explicit.

The correlation-based decoding approach that we have used rests on the idea that if a brain region carries information about the two experimental conditions under consideration, then it should produce repeatable responses within condition, and repeatably *different* responses between conditions. To test this within a scenario, we must inevitably divide the data into even and odd runs to check for repeatability. However, for comparisons across scenarios, we are checking if similar conditions evoke similar response patterns and hence, we do not have to split the data into even and odd runs. Our hypothesis makes no prediction about the average magnitude of the correlations in the within-scenario (Figure 2B) versus between-scenario (Figure 2D) conditions. Thus, in both cases, we used the available data in a way that gave us maximum statistical power to make our inferences.

10. The brief discussion of the generalization effect (Animals-People to Physical-People) in LOC is not very satisfactory. For instance, what kind of similar shape information could be driving this effect that is not present in the other generalization effects? Also, from Table 1 there seems to be a significantly negative generalization effect in V1 for Physical-Objects to Animals-People, but this is not discussed at all.

The pattern of responses correlations in LOC is not crazy and might reflect shape features shared between the two scenarios and/or a general representation of peril. Because the particular features that apparently support this generalization are not critical for any of the hypotheses in this paper, and the effect was not predicted and is only marginally significant, we prefer not to speculate further about them.

On the other hand, the higher correlation for between than within stability in V1 across Physical Objects to the Animals people condition simply makes no sense on any story. We have shown results for all comparisons in all ROIs in the Table, and with 30 such comparisons it is not that surprising that one of them would be a fluke.

11. "We reasoned that if the candidate physics regions are engaged automatically in simulating what will happen next, they should show a higher mean response when viewing physically unstable scenes (because there is more to simulate) than stable scenes (where nothing is predicted to happen)." It seems true enough that, once one knows that a scene is stable, one doesn't then need a dynamically updated representation of its unfolding. But the question that this paper is about is how we determine, in the first place, that a scene is stable or not. The simulations at issue are simulations one runs before one knows their outcome. Stable scenes may well have a lot to simulate, even if we determine after those hefty simulations that the scene is stable after all. And of course unstable scenes might well have very little to simulate, if the scene is simple and the instability is straightforwardly evident. Can the authors say more about why it's easier to determine that a stable scene is stable than that an unstable scene is unstable?

The idea here is that forward simulation happens in all cases but stops if no change has occurred since the last frame. That stopping, both represents the stability of the configuration and produces less activity. This idea is akin to the “sleep state” used for nonmoving objects in a physics engine: they do not need to be re-simulated or re-rendered if they have not moved since the last frame (Ullman et al., 2017 TICS).

12. "Interestingness" ratings seem like a not-quite-adequate approach for evaluating how attention-grabbing the towers were. Why not a measure like how much they distract from another task? The authors could add a new measure of attention-grabbing-ness with a sounder basis in the visual psychophysics literature, or a stronger justification for why "interestingness" ratings are up for the job.

We agreed that it would be useful to obtain such measures, so we spent considerable time and effort to devise a paradigm to measure the distracting effect of attentionally engaging natural images on another task. For these efforts we used the high-arousal and low arousal stimuli from the well validated emotionally arousing “IAPS” stimulus set. None of the many paradigms we tried showed any difference in reaction time when high arousal versus low arousal stimuli were presented during the (orthogonal) main task. We tried placing the task stimuli (Landolt Cs, letters, etc) overlapping on top of the IAPS stimuli, or in the periphery with the IAPS stimuli at the fovea. None of it produced an effect. In looking into the literature, we were unable to find any paradigms that measure effects of the interest level of naturalistic images on secondary tasks. We assume others have tried this before, so the apparent absence of such reports in the literature suggests it simply does not work, presumably because it requires some degree of attention to the stimulus to process its valence.

We note however that we believe our interestingness ratings because they show the predicted ratings for high arousal and low arousal images from IAPS images (which serve as a positive control for these ratings). Thus, we are confident that our interestingness rating paradigm can capture the degree of arousal/attentiveness elicited by images in our dataset.

13. More examples of the stimuli used in each scenario should be provided, either as a supplemental figure or in a hyperlink, to help readers evaluate the stimuli.

Thanks for the suggestion. All stimuli will be shared publicly along with the fMRI data.

[Editors' note: further revisions were suggested prior to acceptance, as described below.]

1. The response to previous point 6, regarding the generalization of the CNN classification, is not completely satisfactory. It's still a worry that the paper is conflating (1) whether CNNs can learn to discriminate stable from unstable with (2) whether whatever it is CNNs do learn is generalizable. The issue raised earlier was that CNN performance was so low that, for all we know, better performance would have revealed that the CNNs' generalizability wasn't so bad after all. To say that the poor performance "reinforces our point that pattern recognition systems are not very good at discerning physical stability" still feels a bit off-target. There are multiple ways to not be good at something, and the authors here are claiming that one of those ways is a lack of generalizability per se. But our previous point was that we can't tell the difference between bad generalizability and bad overall performance. If the authors themselves are making this distinction (and they are in the paper), it seems crucial to separate those claims and support them independently. If they cannot be supported directly from the data, it would be appropriate to discuss this as a limitation of the study.

Our hypothesis was not that *no* CNN could discriminate stable from unstable stimuli, but rather that CNNs trained specifically on object recognition (using ImageNet) could not do this. To test that hypothesis, we asked whether CNN representations that are useful for object recognition can also support discrimination of physical stability in natural scenes. We found low performance both within a scenario type, and a lack of generalizable representation of physical stability in these CNNs. This indicated to us that regions in the brain that are ‘optimized’ for object recognition (i.e., the ventral visual pathway) may also fail to represent physical stability in a generalizable manner. We then tested this hypothesis using fMRI.

We tried to make it clear in multiple places in the manuscript (detailed next) that our hypothesis concerns representations supporting feedforward visual object recognition both in machines (CNNs trained on ImageNet) and brains (ventral visual pathway). But if the reviewer would like to advise us where the confusion arose, we would be happy to try to make it yet clearer.

– [Introduction] “Here we test whether representations that support fast feedforward visual object and scene recognition in machines (i.e., CNNs trained on ImageNet) and brains (i.e., the ventral visual pathway) can support discrimination of the physical stability of complex natural scenes,…”

– [Discussion] “Here we report that fronto-parietal cortical regions previously implicated in intuitive physical inference contain abstract information about physical stability, but representations in the ventral visual object recognition pathway, and those in feedforward convolutional neural networks trained on object recognition, do not. These results indicate that representations in systems that are highly effective at invariant object recognition do not automatically support the general ability to distinguish physically stable from unstable scenes.”

As we now note in the manuscript (Discussion), we do not address here the question of whether a feedforward CNN extensively trained on naturalistic images of stability may learn representations that could discriminate physical stability across multiple scenarios. This is an important question but one we cannot satisfactorily test currently due to lack of large-scale naturalistic datasets of physical stability.

2. The response to previous point 3, regarding the specificity of decoding of representations in the Physics network, did not fully address the issue raised. The authors infer specificity from a null effect in 'decoding' stability/peril in the Animal-People condition, but of course a null effect is not evidence for absence. It would be more appropriate to test whether the decoding in the Physical-Object and Physical-People conditions is statistically better than in the Animal-People condition. E.g. in an ANOVA with two factors: Physical stability vs Animate stability and Within vs. Between. In other words, in their reply the authors cited two findings, (a) a null effect in the Animals-People condition and (b) a significant effect in the Physics condition, but to support their claims it would be useful to test whether b is significantly larger than a. Otherwise, the findings could for instance be due to the Animals-People null effect being based on very noisy measurements, i.e. a non-informative null effect, which does not constitute evidence for the specificity of the physical stability effects.

Thanks for the clarification. We did in fact report the suggested ANOVA results –

“The Physics Network showed the same pattern of results (Figure 2B and Table 1), with significantly higher correlations within than between stability for both the Physical-Object and Physical-People scenarios but not the Animals-People scenarios, and a significant ANOVA interaction between scenario type and within versus between correlations (F_1,12_ = 22.9, p = 0.0004).”